# New Insights into Human Cytomegalovirus pUL52 Structure

**DOI:** 10.3390/v13081638

**Published:** 2021-08-18

**Authors:** Clotilde Muller, Sophie Alain, Claire Gourin, Thomas F. Baumert, Gaëtan Ligat, Sébastien Hantz

**Affiliations:** 1INSERM, CHU Limoges, University of Limoges, RESINFIT, U1092, F-87000 Limoges, France; clotilde.muller@unilim.fr (C.M.); sophie.alain@unilim.fr (S.A.); claire.gourin@etu.unilim.fr (C.G.); 2CHU Limoges, Laboratoire de Bactériologie-Virologie-Hygiène, National Reference Center for Herpesviruses (NRCHV), F-87000 Limoges, France; 3Institut de Recherche sur les Maladies Virales et Hépatiques, Université de Strasbourg, 67000 Strasbourg, France; thomas.baumert@unistra.fr

**Keywords:** human cytomegalovirus, terminase, DNA-packaging, pUL52, viral targets

## Abstract

Human cytomegalovirus (HCMV) can cause serious diseases in immunocompromised patients. Current antiviral inhibitors all target the viral DNA polymerase. They have adverse effects, and prolonged treatment can select for drug resistance mutations. Thus, new drugs targeting other stages of replication are an urgent need. The terminase complex (pUL56–pUL89–pUL51) is highly specific, has no counterpart in the human organism, and thus represents a target of choice for new antivirals development. This complex is required for DNA processing and packaging. pUL52 was shown to be essential for the cleavage of concatemeric HCMV DNA and crucial for viral replication, but its functional domains are not yet identified. Polymorphism analysis was performed by sequencing *UL52* from 61 HCMV naive strains and from 14 HCMV strains from patients treated with letermovir. Using sequence alignment and homology modeling, we identified conserved regions and potential functional motifs within the pUL52 sequence. Recombinant viruses were generated with specific serine or alanine substitutions in these putative patterns. Within conserved regions, we identified residues essential for viral replication probably involved in CXXC-like or zinc finger motifs. These results suggest that they are essential for pUL52 structure/function. Thus, these patterns represent potential targets for the development of new antivirals.

## 1. Introduction

Human cytomegalovirus (HCMV) is a beta-herpesvirus responsible for significant morbidity and mortality in immunocompromised patients and is the leading cause of congenital viral infection [1]. The drugs currently approved against HCMV infection are DNA polymerase (pUL54) inhibitors, including ganciclovir and its prodrug, valganciclovir, cidofovir and foscarnet. Despite their efficiency, the limitations of these drugs are their dose-limiting toxicity and resistances emergence leading to therapeutic challenges [2,3,4,5]. New anti-HCMV molecules coming from closer knowledge of novel targets are hence needed for use instead of, or in combination with, current polymerase pUL54 inhibitors. Interestingly, the letermovir received European and U.S. approval for prophylactic use in hematopoietic stem cell transplants in 2017 [6]. The letermovir acts via a novel, not fully understood, mechanism involving at least the viral terminase proteins pUL56, pUL89 and pUL51 and therefore offers a different mode of action. However, resistance mutations were already characterized in vitro and in vivo [7,8,9].

The terminase complex (pUL56–pUL89–pUL51) is an essential component of the DNA packaging process, which aims to translocate a unit of viral DNA genome into an empty capsid [10]. At least three additional proteins, pUL52, pUL77 and pUL93 seem to be part of the terminase complex and/or participate in the DNA cleavage/packaging process [11,12,13,14,15]. pUL52 is the gene product of ORF *UL52* located on the unique long portion of the viral genome and is essential for viral replication [12]. pUL52 is composed of 668 amino acids with a molecular mass found at 75KDa and is conserved among the members of the Herpesvirus family. To date, pUL32, the homologous protein in Herpes Simplex type 1 (HSV-1), has been further studied. pUL32 is essential for HSV-1 DNA-packaging, and the generation of viral progeny and its localization in cellular compartments seems to depend on the time of infection. Indeed, at 6 to 9 h post-infection (hpi), the protein pUL32 accumulates in viral replication compartments, while later, at 24 hpi, its localization becomes diffused within the nucleus and the cytoplasm [16]. It was shown by a zinc blot assay that pUL32 binds zinc, but amino acids involved in this binding have not been demonstrated so far [17,18]. However, motifs involving cysteines were shown to be essential for viral replication. This kind of motif could bind zinc. The pUL32 results suggest that this protein may be a chaperone protein that modulates disulfide bond formation during procapsid assembly and maturation [16].

A first study conducted by Borst in 2008 showed that (i) pUL52 does not impact viral DNA replication; (ii) pUL52 is required for the cleavage of concatemeric HCMV DNA; (iii) pUL52 is not essential for capsid formation but is essential for obtaining C capsids; (iiii) pUL52 does not seem to be involved in the nuclear localization of the terminase complex [12]. The pUL52 localization studies did not reveal a dynamic localization pattern. pUL52 was localized around the compartments containing pUL56, with some colocalization being apparent at the boundaries. pUL56 is known to be targeted to replication compartments late in the HCMV infection cycle [19]. Thus, in infected cells, pUL52 encloses the nuclear compartments in which replication and packaging of HCMV genomes take place.

The terminase complex is highly HCMV specific, with no counterpart in mammalian cells, and thus represents a target of choice for the development of new antivirals. An essential role of terminase proteins is assumed; nevertheless, their biological functions remain poorly characterized, and their three-dimensional structure has not yet been resolved. Because knowledge of pUL52 functional domains and interaction sites is essential for the future development of drugs targeting the DNA packaging stage, we applied in this work a primary structure analysis to the full-length pUL52 sequence.

## 2. Materials and Methods

### 2.1. Identification of Conserved Patterns and Structure Prediction

The pUL52 amino acids sequences of the HCMV reference strain AD169 strain were aligned with the sequences of 17 homologous proteins from other herpesviruses, as described in Appendix A. Alignments were performed using Clustal Omega multiple sequence alignment (MSA) tools provided by the EMBL-EBI bioinformatics web and programmatic tools framework [20,21,22]. The degree of amino acid conservation is represented as follows: “*” indicates a strictly conserved amino acid; “:” a site belonging to a group exhibiting strong similarity; “.” a site belonging to a group exhibiting a weak similarity. User-identified structural templates selected from the Protein Data Base (PDB) were used to construct full-length atomic models by iterative template fragment assembly simulations (I-TASSER Protein Structure and Function Prediction program) [23,24,25]. Target function was predicted by threading the 3D model through a protein function database. The amino acid sequence of pUL52 HCMV (DAA00157.1) was submitted, and the final model with the higher confidence score (c-score) was analyzed using VMD 1.9.4 (Visual Molecular Dynamics).

### 2.2. Cells and Bacterial Strains

Human fibroblasts MRC-5 (bioMérieux, Craponne, France) were cultivated at 37 °C in 5% CO_2_ and grown in minimum essential medium (MEM) supplemented with 10% fetal bovine serum, 50 μg/mL penicillin and 10 μg/mL gentamycin. *E. coli* strain GS1783 was used for BAC mutagenesis [26]. The HCMV-BAC contained an enhanced green fluorescent protein (EGFP) gene in the unique short region and was derived from parental strain pHB5, the BAC-cloned genome of the HCMV laboratory strain AD169 [27].

### 2.3. HCMV Strains and Isolates

Five reference strains—AD169 (ATCC VR-538), Davis (ATCC VR-807), Towne (ATCC VR-977) Merlin and Toledo—and 75 HCMV clinical isolates collected from various hospitals in France for the National Reference Center for Herpesviruses were studied. Clinical isolates were from congenital HCMV infected newborns or transplanted patients with HCMV infection. Fourteen of the 75 patients were immunocompromised patients that received letermovir as prophylaxis.

### 2.4. Amplification and Sequencing of the UL52 Gene from Reference Strains and Isolates

The full-length *UL52* gene was amplified after DNA extraction (E-Mag, bioMérieux Craponne, France) from isolates by nested PCR using external and internal primers (Appendix A). External PCR and internal PCR were both composed of 500ng of DNA (5µL of external PCR product for internal PCR), 25 µL of the 2X Prime START^®^ Max DNA polymerase (Takara, Maebashi, Japan), 0.3 nM of primers Ext1/Ext2 or Int1/Int2 (described in Appendix A) qsp 50 µL H_2_O PCR grade. The protocol was as follows: an initial denaturation for 5 min at 98 °C then a denaturation for 1 min at 98 °C; 10 s at 98 °C followed by 15 s at 55 °C and 2 min at 72 °C for 45 cycles; a final elongation step for 5 min at 72 °C. After purification of the PCR products using the Wizard^®^ SV Gel and PCR Clean-Up System (Promega, Madison, WI, USA) according to the manufacturer’s instructions, internal PCR products were directly sequenced using the ABI Prism BigDye^®^ Terminator v3.1 Cycle Sequencing Kit in an ABI Prism 3100 Genetic Analyser (Applied Biosystems, Villebon-sur-Yvette, France) with nine sequencing primers designed using Geneious 9.1.8 software. Their specificity was assessed by comparison with GenBank^®^. The sequencing results were analyzed using the Geneious 9.1.8 software by comparison with the AD169 *UL52* sequence.

### 2.5. Heatmap Generation

The heatmap provides a data matrix where the coloring gives an overview of mutational differences between strains. Python 3.9.6 engine is used to generate this binary heatmap. The strains groups are extracted from the base dataset, and the nature of the mutation is recorded in a dummy variable. The base dataset is converted into a two-dimensional matrix (x,y) with y the “strain” ordered by group and x the “amino acids” ordered by “amino acids” id. The binary heatmap is plotted to show mutations in saddle brown and the absence of a mutation in peach puff.

### 2.6. pUL32 HSV-1 Polymorphism Study

Forty-four HSV-1 strains were identified through a sequence similarity search against the HSV-1 pUL32 protein with the NCBI BLAST program. FASTA sequences were analyzed with Geneious 9.1.8 using the Clustal Omega alignment option, and mutations were referenced in an Excel table.

### 2.7. Bacterial Artificial Chromosome Mutagenesis

To identify the crucial amino acids implied in a putative motif, highly conserved residues were substituted with a serine or an alanine by *“en passant”* mutagenesis, using a two-step markerless red-recombination system for BAC mutagenesis in *E. coli* strain GS1783. Single *UL52* mutations were introduced into an EGFP-expressing HCMV-BAC [26], yielding several mutants, as described in Appendix A. Primers used for mutagenesis are described in Appendix A. The presence of the introduced mutation in the *UL52* gene was confirmed for each mutant virus by sequencing prior to transfection. We previously showed that *“en passant”* mutagenesis does not introduce other mutations that could have a negative impact on viral replication [28].

### 2.8. Reconstitution of HCMV-BAC Viruses Harboring the Mutations

Recombinant HCMV-BAC were purified using NucleoBond™ Xtra Midi system (Macherey-Nagel™, Düren, Germany) following the manufacturer’s instructions. To reconstitute virus mutants, purified recombinant BAC were transfected into MRC-5 cells (bioMérieux, France) with the liposomal reagent Transfast™ (Promega, USA) following the manufacturer’s instructions. At 11 days post-transfection, transfected cells were transferred on an MRC-5 culture flask to check the impact of each mutation on viral growth. The presence of mutations in the *UL52* gene of each recombinant virus growing in culture was confirmed by sequencing after viral DNA extraction of each strain using the procedure described by Hirt [29].

### 2.9. Plaque Assays and Growth Curve Analysis

The assessment of the impact of each mutation on viral fitness was performed as previously described [9,28,30]. We inoculated viral recombinant strains and HCMV-BAC AD169 WT in a 48-well MRC-5 culture with a multiplicity of infection (MOI) of 0.01. From day 1 to day 7 post-inoculation, the number of fluorescent cytopathic foci was counted to establish viral growth curves for each recombinant. The absence of contamination was assessed by negative controls and nucleotides polymorphism analysis. The curves represent the average of three independent experiments. For statistical analysis, the Mann–Whitney test was applied. * *p* < 0.05, ** *p* <0.01, *** *p* < 0.001.

### 2.10. Viral Immediate–Early and Late Proteins Expression

Mutated HCMV-BAC were transfected into MRC-5 human fibroblasts using the liposomal reagent Transfast™ (Promega, USA) in a 24 well cell culture plate. At day 5 post-transfection, cells were fixed, and immunostaining was performed for viral immediate–early (anti-IE1 antibody; Argene, Verniolle, France) and late (anti-gB antibody; Abcam, Cambridge, UK) proteins in transfected cells.

## 3. Results

### 3.1. Identification of Conserved Regions

Among the 18 herpesviruses studied (Appendix A), 23 amino acids were strictly similar, 36 strongly similar and 17 were weakly similar. Sequence alignment in the N-terminal part and in the middle part of pUL52 show a lot of gaps and variability between residues, which is characteristic of a variable region. On the contrary, three regions (numbered I to III) from the residue 200 to 282, 451 to 500 and 555 to 591 stand out for the consequent number of retained residues (Figure 1). Of the 27 cysteines present in pUL52 (4.04%), 16 are in a conserved region and 10 of them are highly conserved. Six cysteines highly conserved are found two amino acid residues apart (CXXC motifs). Two of these motifs (C_200_X_2_C_203_ and C_226_X_2_C_229_) are in the conserved region I. C_459_X_2_C_462_ is in the conserved region II. Another pattern highly conserved involving one histidine and two cysteines is found in region III (H_495_X_2_C_498_X_3_C_502_). The I-TASSER report indicates a potential ligand-binding site that implicates cysteine and histidine in the region that we found conserved.

### 3.2. Identification of Sequence Polymorphisms

The pUL52 sequence of reference strains AD169, Towne and Toledo were identical to the sequences found in GenBank (accession number FJ527563.1, GQ121041.1, GU937742), and inter-assay reproducibility was 100%. Among the 61 naïve strains sequenced, we described 36 amino acid polymorphisms distributed across pUL52. Seven are also present in the reference strains Towne, Toledo, Davis or Merlin. The mean identity of the HCMV isolates was 99.4% (range: 98.2–100%), and the mean number of polymorphisms per isolates was 3.4 (range: 0 to 12). Analysis of polymorphism repartition through pUL52 revealed two parts with a huge concentration of mutations which are consistent with the alignment where gaps and variability were found. These two variable regions are annotated as pUL52 VRI and VRII. Of the 61 naive strains studied, mutations do not affect areas identified as functional. Only two mutations are found once in a conserved region: V482L in region II (451 to 500) and V580I in region III (555 to 591). Sequence alignment showed that these two residues are not conserved among herpesviruses. Conserved and variable regions identified by alignment can overlay with the natural polymorphism identified in pUL52 isolates (Figure 1). In the sequences of the 14 strains from patients treated with letermovir, 18 mutations were found: seven were common to reference strains, five to naive strains and six never described (L66F; G115V; E335Q; S392P; del408; H420N). L66F and G115V are in the variable region I and S392P, del408, H420N in the variable region II. The residue E335 is close to the putative NLS described by Borst [12]. However, this amino acid is not conserved between herpesviruses and beta-herpesviruses, and none of these mutations were found in one of the three regions determined as conserved.

### 3.3. pUL32 Polymorphism Is Consistent with pUL52

The polymorphism of the HSV-1 pUL32 protein, the HCMV pUL52 homolog, was studied from 44 different strains found in Genbank^®^: 42 mutations were found through the 596 amino acids that composed pUL32. The polymorphism distribution reveals a variable region from amino acid 0 to 100 and a second longer variable region from amino acid 250 to 390 (Figure 2a). These two variable regions are consistent with the regions described in pUL52. Four motifs, _128_CXXC_131_, _155_CXXC_158_, _423_CXXC_426_ and _495_HXXCXXXC_502,_ described by Albright et al., are not in polymorphism-rich regions and are conserved between the three Herpesvirus subfamilies [16]. Only the _291_CXXXXC_297_ motif is located in a polymorphism-rich region that we called VRII. This motif is only found in alpha-herpesviruses (Figure 2b).

### 3.4. Analysis of Conserved Region I Revealed Putative Metal-Binding Sites

The alignment of the conserved region I with 17 homologs from herpesviruses showed that, over the length, 17 residues are conserved. Nine of them are perfectly conserved (C200, C203, Y222, C226, D272, H276, F277, C282, F283), and eight are semi-conserved, mainly nonpolar residues (5 aliphatics: V/L/I202, V/L/I206, L/I225, V/L/I259, V/L/I268; one aromatic W/F217; two polar negatively charged D/E220, D/E245). In this high conserved region rich in cysteines, two CXXC motifs are found 22 amino acid residues apart: _200_CXXC_203_ and _226_CXXC_229_. A region between these motifs is particularly conserved from amino acid 217 to 222 with a W/F-X-X-D/E-Y type sequence (Figure 3b). We are in the presence of a cyclic amino acid separated by two negatively charged residues, directly followed by a tyrosine. The amino acids sequence of pUL52 HCMV was submitted in I-TASSER to further investigate the pUL52 theoretical structure. The predictive secondary structure of the conserved region I is composed of a helix with a high confidence score. To better visualized the motif, the conserved region is selected with VMD 1.9.4. The shape of this region is typical of a zinc finger, cysteine C200, C203, C226 and C229 could be implicated in this shape, and the conserved amino acids 217 to 222 may constitute the binding site of the zinc finger. (Figure 3c). The strictly conserved residues H276 and C282 are very close to the four cysteines C200, C203, C226 and C229 and could also be implicated. I-TASSER data considered this motif as a predictive zinc-binding site. To further investigate the implication of these residues in a putative zinc-finger motif, we produced by “*en passant*” mutagenesis recombinant EGFP-viruses (C200S, C203S, C226S, C229S, H276A, C282S). Eleven days after transfection in human fibroblasts, no cytopathogenic effect was observed for mutations C200S, C203S, C226S, H276A and C282S located within the putative metal-binding site (Figure 3d, Appendix A). The HCMV-BAC *UL52* C229S can replicate in human fibroblasts. However, C229S mutation significantly reduced the virus capacity to produce infectious particles, showing that C229 is an important amino acid for HCMV replication fitness (Figure 3e).

### 3.5. Analysis of Conserved Region II

The conserved region II is composed of 49 residues from 451 to 500 (Figure 4a). Four residues are strictly conserved, eight are strongly conserved and three are weakly. Of the four residues strictly conserved, two cysteines are found two residues apart (Figure 4b). It corresponds to a CXXC motif. This _459_CXXC_462_ motif is consistent with the _423_CXXC_426_ motif of pUL32. HCMV-BAC-*UL52*-C459S and HCMV-BAC-*UL52*-C462S were produced, and eleven days after transfection in human fibroblasts, no cytopathogenic effect was observed for these two mutations, indicating the importance of this motif for viral replication and thus for protein function (Figure 4c). A mutation of region II found in one of the 61 naïve strains (V482L) is not conserved between herpesviruses studied (Figure 4b, Appendix A).

### 3.6. Analysis of Conserved Region III

The conserved region III ranges from residue 555 to residue 591 (Figure 5a). This high conserved region is composed of six amino acids strictly conserved (K566, H567, F569, D571, C574, L590), three strongly (L555, F568, F591) and two weakly (I564, Y565). A motif involving H567, C570 and C574 is like those previously described in pUL32 _495_HXXCXXXC_502_ and predicted in alpha-helix (Figure 5b,c). Only the RCMV-pUL52 and MCMV-pUL52 do not have this motif strictly conserved in the other 17 homologs (Figure 5b). These two homologs have a threonine or an alanine instead of the cysteine present in the other herpesviruses studied. At 11 days post-transfection of the HCMV-BAC-*UL52*-C567S and HCMV-BAC-*UL52*-C574S, no foci were observed indicating the importance of these residues (Figure 5d, Appendix A). The HCMV-BAC-*UL52*-C570S did not show foci at day 11 but the passage in a flask culture allowed to observe a slight growth. The virus growth was slow, showing that C570 has an impact on HCMV replication fitness (Figure 5e). A mutation of region III found in one of the 61 naïve strains (V580I) is not conserved among herpesviruses studied (Figure 5b).

### 3.7. pUL52 Mutations Do Not Impact the Viral Gene Expression

Expression of immediate–early and late viral genes were detected by immunostaining 5 days after transfection in human fibroblast, indicating that the different recombinant viruses built for testing the impact of amino acids mutations on viral replication did not impact the expression of the viral genes (Appendix A).

## 4. Discussion

HCMV is responsible for significant morbidity and mortality in immunocompromised patients and in congenitally infected neonates. pUL52 was shown to be essential for the cleavage of concatemeric HCMV DNA and crucial for viral replication. To date, its functional domains were not identified.

Taken together, our results confirmed that the amino acids sequence of pUL52 is highly conserved among HCMV reference strains and clinical isolates, with an average sequences identity of 99.4%. Comparison of the HCMV pUL52 sequence with its homologs showed three highly conserved regions and two variable regions (Figure 6a). By studying polymorphism, we were able to refine the conserved and variable regions initially described by the sequence alignment. Two regions are rich in polymorphisms. The first one is located in the N-term from amino acid 19 to amino acid 150, and the second is the middle part of the protein from amino acid 330 to 410. They are consistent with the variable regions defined by the alignment of pUL52 homologs. Only two mutations are found in the three conserved regions identified (V482L in region II and V580I region III). These two amino acids are not conserved in herpesviruses, and mutations V to I or V to L do not change the amino acid properties. Within a highly conserved protein, these results suggest that these regions (VRI and VRII) correspond to non-functional regions of pUL52. In contrast, the conservation of regions I to III indicates their likely involvement in specific enzymatic functions of pUL52 or in the conservation of its three-dimensional structure.

Interestingly, analysis of HSV-1 pUL32 sequences from 44 strains retrieved from the Genbank^®^ reveals more diffuse polymorphisms than in HCMV-pUL52. In general, the polymorphism is like that of pUL52, and these results were found with sequence alignment. Indeed, sequence alignment in the N-terminal part and in the middle part showed a lot of gaps and variability between residues. It appears that these two variable regions identified in HSV-1 and HCMV have the same location. It is unlikely that an essential function would be associated with these regions. Four motifs CXXC were described in pUL32; three of them are located in the most conserved regions. One of the four motifs is found in a region rich in polymorphisms (VRII), and this motif is not conserved in beta- and gamma-herpesviruses. In addition, the mutation of *UL32*-C291A does not lead to a loss of viral replication. Thus, the essential character of this motif was not validated to date [16].

Alignment with herpesviruses highlights a large, conserved region that we called conserved region I. A polymorphism study between 75 HCMV strains revealed no mutations in this region. This region is rich in cysteine and histidine (15.6%) and can have zinc-binding sites. In the same way, I-TASSER report indicated the potential implication of the C200, C203, C226, C229 or C200, C203, H276 and C282 in the zinc-binding sites. The shape of this region visualized with VMD 1.9.4 showed a typical shape of the zinc finger with C200, C203, C226, C229. We hypothesize a zinc finger implicating the four cysteines C200, C203, C226, C229. This motif was discovered in various DNA binding proteins such as the T4 phage terminase complex protein: the gp49 protein [31]. To further study the implication of cysteine and histidine, recombinant viruses were built. Only the C229S mutant can replicate, but the viral fitness is impacted compared to the reference strain AD169. This shows the importance of these amino acids for viral replication. We can hypothesize that these mutations affect the function of the protein, probably preventing interaction with viral DNA.

The conserved regions II and III are delimitated as follows: region II from residue 451 to residue 500 and region III from residue 555 to residue 591. According to in silico reconstruction of the whole protein using I-TASSER, region II from amino acid 480 to 500 and region III from amino acid 555 to 591 are close and could be in favor of interactions (Figure 6b). These two high conserved regions are composed of motif implicating cysteines. In region II we found the _459_CXXC_462_ motif, and in region III, the _567_HXXCXXXC_574_ that are consistent with the _423_CXXC_426_ motif and the _495_HXXCXXXC_502_ motif found in HSV-1 pUL32. Substitution of these cysteine/histidine in serine or alanine in an HCMV-BAC-AD169 allowed us to validate the importance of the residues C459, C462, H567 and C574 for viral replication. The HCMV-BAC with *UL52*-C570 serine substitution affects the viral growth. Interestingly, this cysteine is not strictly conserved in other herpesviruses, and the same results are shown with the substitution of the *UL32*-C498 in alanine.

Sequence alignment between the eight beta-herpesviruses showed a particularly conserved C-terminal region from residue 617 to residue 668. This region is rich in amino acids with an amino group contained in the side chain: arginine and lysine. This region is poor in polymorphisms. The mutation S652N is found in the reference strains Merlin and Toledo and most of the studied strains. Another mutation, A640T, is found in one naïve strain. These mutations do not concern conserved amino acids. The first study of pUL52 showed that the C-terminal part of this protein is essential for its function and for its correct localization [12]. They assumed the presence of a putative NLS at positions 315 to 321. Because of the importance of the C-terminal part and given the sequence alignment of this part in beta-herpesviruses, we can assume the presence of a second NLS from the amino acid composition of this region.

Collectively, our study provides new insights into the pUL52 structure and will contribute to a better understanding of the pUL52 function. In addition, the study of pUL52 polymorphisms and conserved domains should facilitate the identification of resistance mutations to the inhibitor of terminase complex when they occur in clinical practice. Finally, our data pave the way for the identification of previously undiscovered antiviral targets and the development of drugs urgently needed.

## Figures and Tables

**Figure 1 viruses-13-01638-f001:**
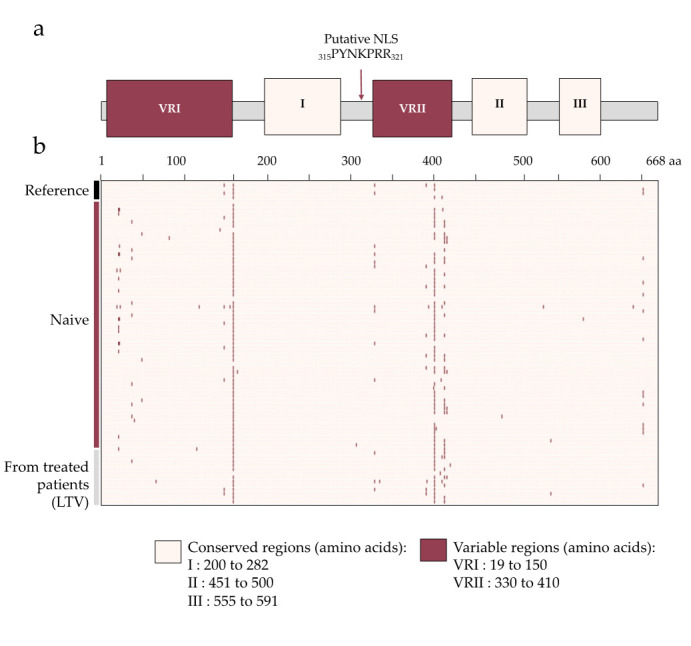
Structure and polymorphism distribution of HCMV pUL52. (**a**) pUL52 is composed of 3 conserved (I–III) and 2 variable regions (VRI and VRII). A putative nuclear localization signal (NLS) of pUL52 is shown by a red arrow at the amino acid position 315 to 321. (**b**) Representation through a binary heatmap of the polymorphism distribution according to the strain and the position of the mutation in pUL52. The 3 groups studied (reference strains (*n* = 5), naïve strains (*n* = 61) and strains from patients treated with letermovir (LTV) (*n* = 14) are represented by black, saddle brown and grey boxes, respectively, in the y axis. Mutations are shown in saddle brown and the absence of a mutation in peach puff. VRI and VRII: variable regions I and II. aa: amino acids.

**Figure 2 viruses-13-01638-f002:**
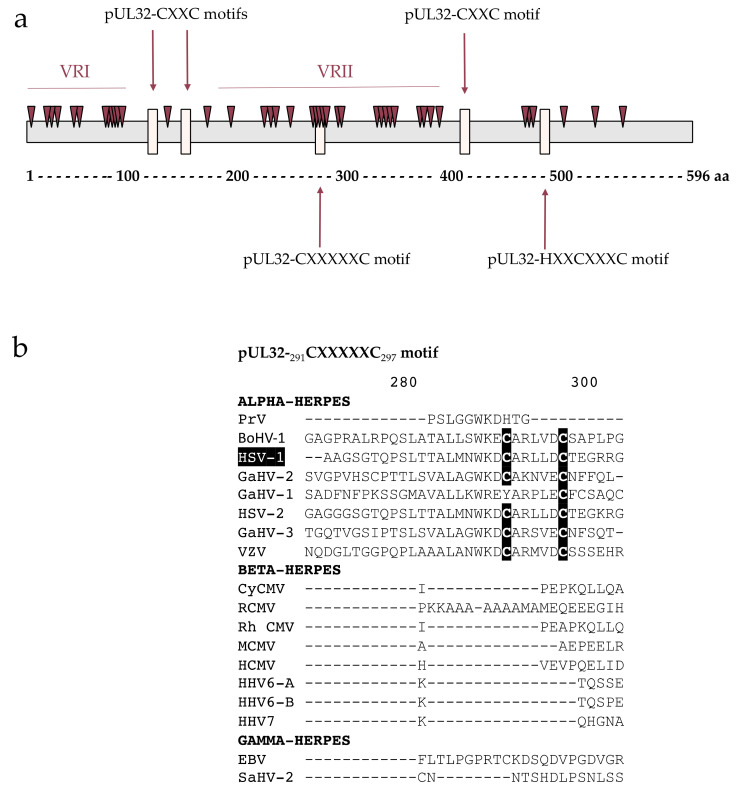
Distribution of the polymorphism through the HSV-1 pUL32 protein and motifs previously described by Albright in 2015. (**a**) pUL32 strains were identified through a sequence similarity search against pUL32 using the NCBI BLAST program. The 44 pUL32 HSV-1 strains were aligned using Clustal W alignment with Geneious 9.1.8 software. Each mutation is annotated as a saddle brown triangle. Motifs previously described are represented with a peach puff rectangle. (**b**) Sequence alignment from 18 herpesviruses of the pUL32-_291_CXXXXXC_297_ motif. Sequence numbering is consistent with that of HSV-1 strain residues. Key residues involved in the CXXXXXC motif are shown as white letters on black background. VRI and VRII: variable regions I and II. aa: amino acids.

**Figure 3 viruses-13-01638-f003:**
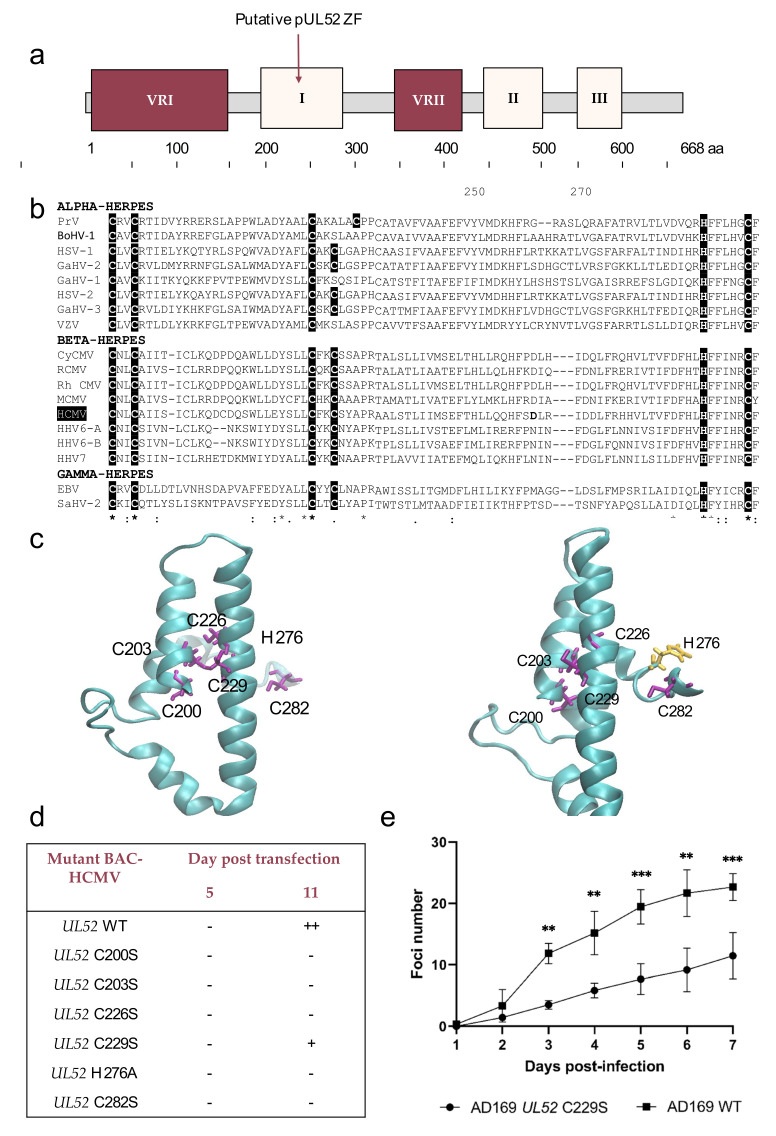
Identification of a putative zinc-finger motif within pUL52 protein. (**a**) Structure of HCMV pUL52 with a putative zinc-finger (ZF) pattern (VRI and VRII: variable regions I and II). (**b**) Sequence alignment of the conserved region I from 18 herpesviruses and residues involved in a metal-binding site. Sequence numbering is consistent with that of HCMV reference strain AD169 residues. Key residues involved in the formation of the zinc-finger motif are shown as white letters on black background. (**c**) The region I was extracted from the pUL52 theoretical structure. Cysteines are represented in purple, histidine H276 in yellow. (**d**) Impact of HCMV-BAC-*UL52* mutants on growth in cell culture at day 11 (fibroblasts MRC-5). (**e**) Growth curves of the recombinant virus strain HCMV-BAC-*UL52*-C229S in comparison to the parental strain HCMV-BAC AD169 WT. Fluorescent foci were counted daily from day 1 to day 7. Curves are the average of three independent experiments. ** *p <* 0.01; *** *p* < 0.001 (Mann–Whitney test). VRI and VRII: variable regions I and II. aa: amino acid.

**Figure 4 viruses-13-01638-f004:**
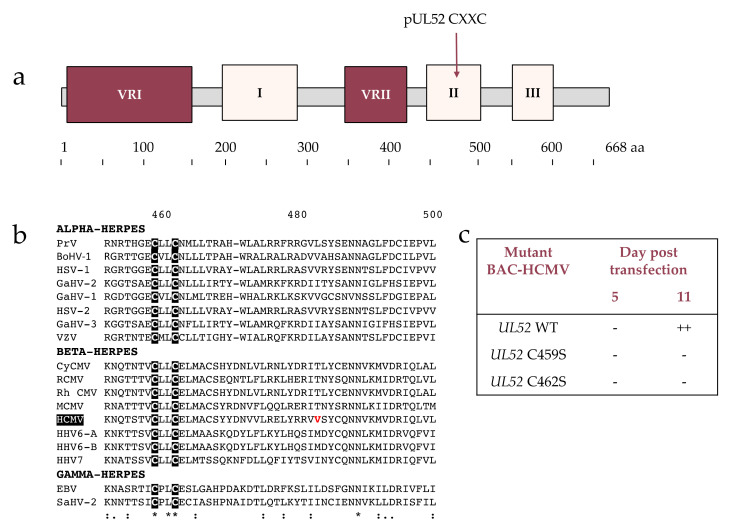
CXXC-like motif in conserved Region II. (**a**) Structure of HCMV pUL52 with a CXXC motif. (**b**) Sequence alignment of the conserved region II from 18 herpesviruses and residues involved in a CXXC motif. Sequence numbering is consistent with that of HCMV reference strain AD169 residues. Key residues involved in the formation of the CXXC motif are shown as white letters on black background. (**c**) Impact of HCMV-BAC-*UL52* mutants on growth in cell culture at day 11 (fibroblasts MRC-5). VRI and VRII: variable regions I and II. aa: amino acid.

**Figure 5 viruses-13-01638-f005:**
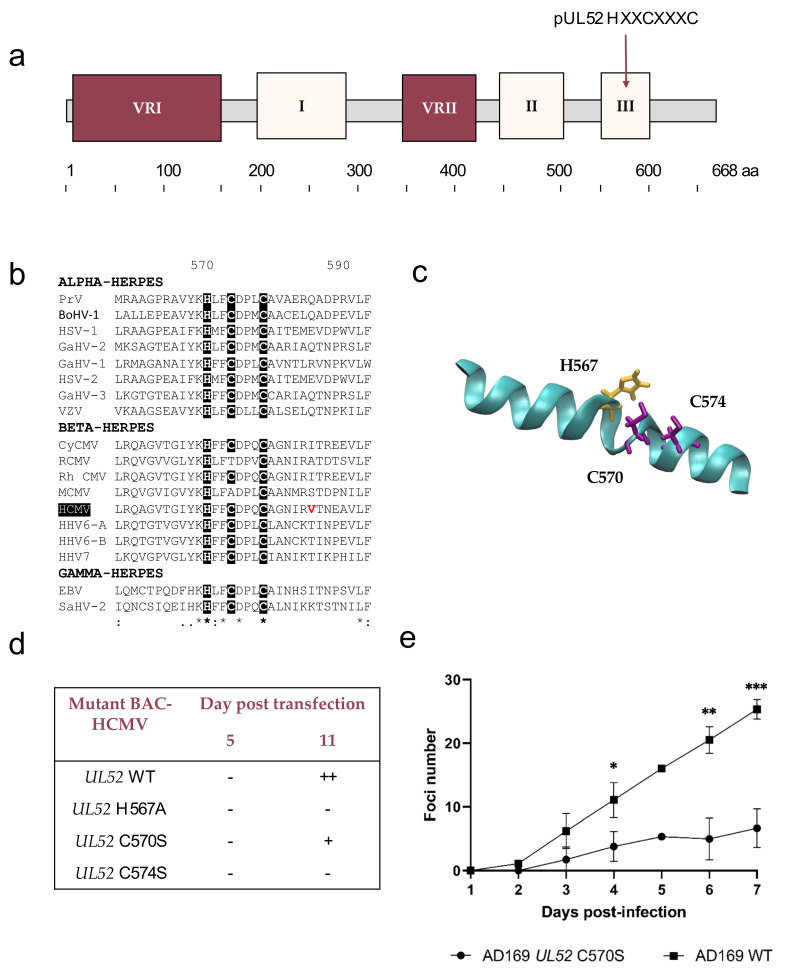
HXXCXXXC motif in conserved Region III. (**a**) Structure of HCMV pUL52 with an HXXCXXXC motif. (**b**) Sequence alignment of the conserved region III from 18 herpesviruses and residues involved in an HXXCXXC motif. Sequence numbering is consistent with that of HCMV reference strain AD169 residues. Key residues involved in the formation of the HXXCXXC motif are shown as white letters on black background. The residue in red matches the position of the mutation V580I found in one of the 61 naive strains. (**c**) Region 3 was extracted from the pUL52 theoretical structure. Cysteines and histidine are represented in purple and yellow, respectively. (**d**) Impact of HCMV-BAC-UL52 mutants on growth in cell culture at day 11 (fibroblasts MRC-5). (**e**) Growth curves of the recombinant virus strain HCMV-BAC-*UL52*-C570S in comparison to the parental strain HCMV-BAC AD169 WT. Fluorescent foci were counted daily from day 1 to day 7. Curves are the average of three independent experiments. * *p* < 0.05; ** *p <* 0.01; *** *p* < 0.001 (Mann–Whitney test). VRI and VRII: variable regions I and II. aa: amino acid.

**Figure 6 viruses-13-01638-f006:**
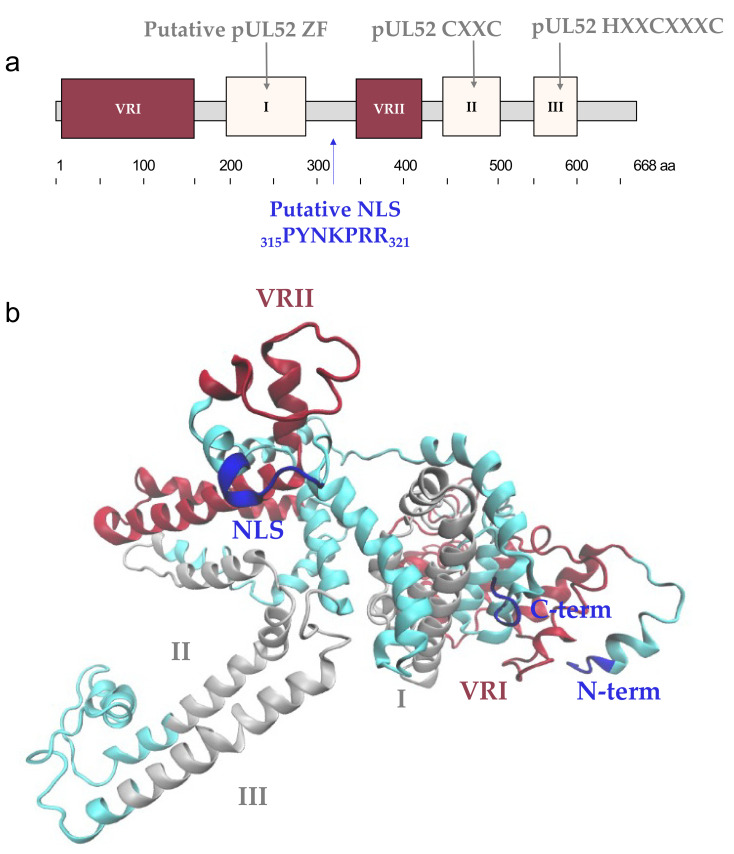
Structure of pUL52. (**a**) Structure of HCMV pUL52 with conserved and variable regions. The putative zinc-finger (ZF), CXXC and HXXCXXXC patterns are in the conserved regions I, II, III, respectively. (**b**) Predictive atomic structure of pUL52. The amino acid sequence of pUL52 HCMV (DAA00157.1) was submitted in I-TASSER (Iterative Template fragment Assembly Simulations), and the final model with the higher confidence score (c-score) was analyzed using VMD 1.9.4 (Visual Molecular Dynamics). Variable and conserved regions determined by polymorphism and sequence alignment analysis are represented in saddle brown and grey, respectively. The putative NLS _315_PYNKPRR_321_, N-term and C-term are represented in blue. VRI and VRII: variable regions I and II. aa: amino acid.

## Data Availability

The data are available from the corresponding authors upon reasonable request.

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
