# Peer review of "New Insights into Human Cytomegalovirus pUL52 Structure"

_viruses, 2021, doi:10.3390/v13081638_

Round 1

Reviewer 1 Report

The manuscript by Muller et al. deals with the analysis of functional domains of HCMV pUL52. By using naive and patient derived HCMV strains, the authors identified conserved regions of pUL52. The in silico data were confirmed by construction of recombinant viruses. The authors found two motifs: a CXXC-like and a zinc finder motif. Overall, the manuscript is well described, convincing and lead to further insights into the function of this essential packaging protein.

Specific points:

  1. 157 and 159 Please delete one sentence
  2. Concerning growth curve analysis: include a brief description in Material and Methods
  3. Until recombinant viruses C229S and others are able to replicate, this reviewer recommend a plaque reduction assay (yield assay). This should give further insides into the function of these mutants. The read out should be PFU/mL
  4. An in silico reconstruction of the whole protein would be helpful to see if there will be neighbouring aa that effect interactions
  5. Line 67: the first detailed description of the NLS of pUL56 is described in Giesen et al., J.Gen.Virol. 2000,Vol 81

Author Response

We would like to thank the editor and the reviewers for giving us the opportunity to revise our manuscript. We carefully addressed the comments of the reviewers.

Point by point response to Reviewer 1

The manuscript by Muller et al. deals with the analysis of functional domains of HCMV pUL52. By using naive and patient derived HCMV strains, the authors identified conserved regions of pUL52. The in silico data were confirmed by construction of recombinant viruses. The authors found two motifs: a CXXC-like and a zinc finder motif. Overall, the manuscript is well described, convincing and lead to further insights into the function of this essential packaging protein.

Specific points:

1- 157 and 159 Please delete one sentence.

We apologize for this mistake. We agree with the reviewer that this should be modified. We deleted the sentence in the manuscript (page 4, line 157).

2- Concerning growth curve analysis: include a brief description in Material and Methods

We thank the reviewer for this comment. We added a description in the 2.8 section.

Page 4, line 155-160: “We inoculated viral recombinant strains and HCMV-BAC AD169 WT in 48-well MRC-5 culture with a multiplicity of infection (MOI) of 0.01. From day 1 to day 7 post-inoculation, the number of fluorescent cytopathic foci was counted to establish viral growth curves for each recombinant.”

3- Until recombinant viruses C229S and others are able to replicate, this reviewer recommend a plaque reduction assay (yield assay). This should give further insides into the function of these mutants. The read out should be PFU/mL

As shown in Figure 3e and 5e, to estimate the fitness impact of mutations C229S and C570S on virus replicative capacity, we then compared the growth curves of the wildtype and mutant viruses. Both mutants (C229S and C570S) grew more slowly than the wildtype virus

4- An in silico reconstruction of the whole protein would be helpful to see if there will be neighbouring aa that effect interactions

We thank the reviewer for this very relevant suggestion. We added an additional figure comprising the theoretical HCMV pUL52 protein structure. The figure 6 and the legend are added at the end of the manuscript (page 17 line 485 to 495). The figure 6 a is mentioned page 13 line 323. The figure 6 b is mentioned page 13-14 line 362 to 365 with the following text : “According to in silico reconstruction of the whole protein using I-TASSER, region II from amino acid 480 to 500 and region III from amino acid 555 to 591 are close and could be in favor of interactions (Figure 6b)”.

5- Line 67: the first detailed description of the NLS of pUL56 is described in Giesen et al., J.Gen.Virol. 2000,Vol 81

We agree with the reviewer, The reference was replaced by Giesen et al. 2000.

Reviewer 2 Report

The pUL52 protein is essential for cleavage of HCMV DNA and viral replication, thus better understanding of its functional domains is critical for HCMV study and associated-drug development. This is an interesting study and authors have collected some unique dataset for analyzing the motifs of pUL52 protein. Detailed analysis studies on the mutations in pUL52 proteins, and the viral replications for the mutation ones were confirmed by growth curve.  The whole paper is generally well written and structured. In my opinion, the manuscript is suitable for publishing on Viruses, after the authors modified the following comments:

Please make your figures with higher resolution.

Author Response

We would like to thank the editor and the reviewers for giving us the opportunity to revise our manuscript. We carefully addressed the comments of the reviewers.

Point by point response to Reviewer 2

The pUL52 protein is essential for cleavage of HCMV DNA and viral replication, thus better understanding of its functional domains is critical for HCMV study and associated-drug development. This is an interesting study and authors have collected some unique dataset for analyzing the motifs of pUL52 protein. Detailed analysis studies on the mutations in pUL52 proteins, and the viral replications for the mutation ones were confirmed by growth curve.  The whole paper is generally well written and structured. In my opinion, the manuscript is suitable for publishing on Viruses, after the authors modified the following comments:

Please make your figures with higher resolution.

We thank the reviewer for his comment. To improve the reader’s visibility, figures have been re-integrated into the manuscript with a higher resolution.

Round 2

Reviewer 1 Report

The authors addressed all points of this reviewer. The manuscript impove a lot and is now suitable for publication in Viruses.